# Development and testing of a deep learning algorithm to detect lung consolidation among children with pneumonia using hand-held ultrasound

David Kessler[1]*, Meihua Zhu[2], Cynthia R. Gregory[2], Courosh Mehanian[2,3,4], Jailyn Avila[5], Nick Avitable[1], Di Coneybeare[1], Devjani Das[1], Almaz Dessie[1], Thomas M. Kennedy[1], Joni Rabiner[1], Laurie Malia[1], Lorraine Ng[1], Megan Nye[1], Marc Vindas[1], Peter Weimersheimer[6], Sourabh Kulhare[4], Rachel Millin[4], Kenton Gregory[2], Xinliang Zheng[4], Matthew P. Horning [4], Mike Stone[7], Fen Wang[2,8], Christina Lancioni [2]

1 Department of Emergency Medicine, Columbia University Vagelos College of Physicians & Surgeons, New York Presbyterian Morgan Stanley Children's Hospital, NY, NY, United States of America, 2 Oregon Health & Science University, Portland, Oregon, United States of America, 3 University of Oregon, Eugene, OR, United States of America, 4 Global Health Labs Inc, Bellevue, WA, United States of America, 5 University of Kentucky, Lexington, KY, United States of America, 6 University of Vermont Larner College of Medicine, Burlington, VT, United States of America, 7 Legacy Emanuel Medical Center, Portland, OR, United States of America, 8 Fudan University, Shanghai, China

* dk2592@cumc.columbia.edu

**Data Availability Statement:** the data underlying the findings described within this manuscript are freely available in a public repository: (https://

## Abstract

### Background and objectives

Severe pneumonia is the leading cause of death among young children worldwide, disproportionately impacting children who lack access to advanced diagnostic imaging. Here our objectives were to develop and test the accuracy of an artificial intelligence algorithm for detecting features of pulmonary consolidation on point-of-care lung ultrasounds among hospitalized children.

### Methods

This was a prospective, multicenter center study conducted at academic Emergency Department and Pediatric inpatient or intensive care units between 2018–2020. Pediatric participants from 18 months to 17 years old with suspicion of lower respiratory tract infection were enrolled. Bedside lung ultrasounds were performed using a Philips handheld Lumify C5-2 transducer and standardized protocol to collect video loops from twelve lung zones, and lung features at both the video and frame levels annotated. Data from both affected and unaffected lung fields were split at the participant level into training, tuning, and holdout sets used to train, tune hyperparameters, and test an algorithm for detection of consolidation features. Data collected from adults with lower respiratory tract disease were added to enrich the training set. Algorithm performance at the video level to detect consolidation on lung ultrasound was determined using reference standard diagnosis of positive or negative pneumonia derived from clinical data.

dataverse.harvard.edu/dataset.xhtml?persistentId=
doi:10.7910/DVN/3HP1RD).

**Funding:** This work was supported via a Defense
Advanced Research Projects Agency (DARPA)
award "Hand-Held Convolutional-Neural-Network
based Field Diagnostic Ultrasound" Technology
Investment Agreement No. HR0011-17-3-0001 to
Inventive Government Solutions, LLC and
subcontracted to OHSU. Co-funding was also
provided by the Global Good Fund. There was no
additional external funding received for this study.
The funders provided support in the form of
salaries for authors, but did not have any additional
role in the study design, data collection and
analysis, decision to publish, or preparation of the
manuscript. The specific roles of these authors are
articulated in the 'author contributions' section".

**Competing interests:** Dr. Kenton Gregory received
compensation as Principal Investigator for a
DARPA award to Inventive Government Solutions
Inc, previously owned by Intellectual Ventures
Laboratory. Dr. Kenton Gregory also received
compensation as a Fellow from Intellectual
Ventures Laboratory for a portion of this work. This
potential conflict for Drs. Kenton and Cynthia
Gregory has been reviewed and managed by
OHSU. Since the time this research was
completed, Global Health Labs Inc. acquired all the
assets of Inventive Government Solutions and
Intellectual Ventures Laboratory. Drs. Mehanian
and Millin, as well as S. Kulhare and M. Horning,
received salary support through Global Health Labs
Inc., a non-profit organization that may have a
commercial interest in the results of this research
and technology. Global Health Labs Inc. has been
awarded proprietary ownership of the code utilized
in the algorithm. These commercial affiliations do
not alter our adherence to PLOS ONE policies on
sharing data and materials

## Results

Data from 107 pediatric participants yielded 117 unique exams and contributed 604 positive and 589 negative videos for consolidation that were utilized for the algorithm development process. Overall accuracy for the model for identification and localization of consolidation was 88.5%, with sensitivity 88%, specificity 89%, positive predictive value 89%, and negative predictive value 87%.

## Conclusions

Our algorithm demonstrated high accuracy for identification of consolidation features on pediatric chest ultrasound in children with pneumonia. Automated diagnostic support on an ultraportable point-of-care device has important implications for global health, particularly in austere settings.

## Introduction

Pneumonia is a leading cause of global pediatric morbidity and mortality, accounting for 14% of all deaths of children under five in 2019 [1]. Deaths from pediatric pneumonia are disproportionately seen in South Asia and Sub-Saharan Africa and represent the single largest infectious cause of death in children worldwide [1–3]. Consequently, the World Health Organization offers guidelines for hospital admission and empiric antibiotic treatment for presumptive bacterial pneumonia based solely on clinical examination that have led to decreased mortality in resource limited settings [4]. However, chest imaging is still optimal for improving diagnostic accuracy and delivering precision care [5, 6]. Specifically, clinical features such as tachypnea, chest retractions, and hypoxemia, are shared among distinct etiologies of respiratory tract infections in young children, such as viral bronchiolitis and lobar pneumonia, and have different treatment strategies. There is growing concern that antibiotics are overused in the treatment of children presenting with signs and symptoms of lower respiratory tract infection (LRTI), and this may facilitate emergence of antibiotic-resistance [7]. Moreover, lack of access to chest imaging may overlook key severity features of bacterial LRTI, such as the presence of loculated pleural effusions, that require distinct treatment approaches.

Chest radiograph (CXR) is the current gold standard for evaluation of suspected pneumonia or its complications [8–10]. However, cost, access to technology, lack of timely availability of physicians for image interpretation, and exposure to radiation make CXR a less ideal imaging modality, especially for children in low resource settings. Lack of portable CXR equipment, as well as the inferior quality when available, also limits access to quality chest imaging for children with critical illness who cannot be transported to imaging facilities in low resource settings [11]. Ultrasound is a non-ionizing and less expensive imaging modality that provides comparable or even superior diagnostic accuracy for pneumonia in children [12–16]. Several recent meta-analysis demonstrate that lung ultrasound is reliable for identification of pneumonia in both children and adults [17–19]. In addition, portability of the equipment, battery-power, quick operation, and ease of serial examinations makes this an ideal technology for austere settings or anywhere that diagnostic imaging at the point-of-care would be beneficial [20]. The primary limitation of ultrasound relates to dependency on operator experience for both acquiring and interpreting the images. However, use of simple standardized imaging protocols

to collect data, and artificial intelligence (AI) for image interpretation, offers the potential to close these gaps [21].

Our overall goal is to optimize a pediatric-specific deep-learning algorithm that can be embedded in a point-of-care, hand-held ultrasound device, in order to provide bedside identification of lobar pneumonia, pleural effusion, and empyema among young children presenting with signs and symptoms of LRTI in low resource settings. High sensitivity and specificity, accurate feature localization, and rapid processing time are required attributes of an AI-enabled point-of-care device in order to gain widespread clinical acceptance. Algorithm development is a supervised iterative process of design, training, and fine-tuning to meet these requirements. Our team previously published data on the development and accuracy of a deep-learning algorithm to automatically detect sonographic lung pathology using training data from a swine model, however the model was not transferable to pediatric patient images [22]. In the current study, our primary aim was to develop and test the accuracy of an AI algorithm to detect features of pulmonary consolidation on point-of-care ultrasounds among pediatric patients.

## Methods

### Study population

Data for design and testing of the algorithm was collected from prospectively enrolled pediatric participants (age 18 months to 17 years old) with suspicion of LRTI or associated complications (e.g., pneumonia, pleural effusion, empyema) between 2018–2020, at two major US pediatric academic centers. Patients were excluded if they were intubated, could not tolerate a lung ultrasound (LUS) due to stress or anxiety of child or parent, or if unable to access chest wall to perform a LUS due to surgical dressing or open wound. To identify potentially eligible patients for study inclusion, the study team screened the current inpatient or Emergency Department census for patients with suspected LRTI and following informed consent (and assent when appropriate) had a study LUS performed. Subsequently, data from the medical records from each participant's relevant hospitalization were reviewed to obtain demographic and clinical data and to identify individuals diagnosed with and treated for pneumonia. Specifically, data detailing admission and discharge diagnoses, results of any microbiologic testing (respiratory viral panel results were available for 77% of participants), treatment course, performance of pulmonary procedures (eg, chest tube placement; bronchoscopy), and clinical radiologists' interpretation of chest imaging (available for 89% of participants), were collected. Using this approach, participants with diagnostic studies, clinical course, and documentation supporting a diagnosis of bacterial pneumonia were systematically delineated from participants with other disease processes who were not diagnosed with pneumonia for inclusion in the final training dataset. Adults 18–80 years old with a clinical diagnosis of pneumonia and CXR and/or chest CT evidence of consolidation were enrolled separately, following provision of written informed consent. Adult participants were excluded if they were intubated, had active sepsis, cardiogenic pulmonary edema, known lung cancer, pulmonary embolus, chronic bronchiectasis or cystic fibrosis, or could not tolerate the LUS procedure.

### Ethics approval and consent to participate

Written, informed consent was provided by the parent or legal guardian for each participant with assent from children seven years or older in accordance with approval from each sites' Institutional Review Board. All adult participants also provided written informed consent in accordance with recruitment site Institutional Review Boards.

## Scanning protocol and imaging data

All imaging was performed with Philips handheld C5-2 transducers using the Lumify system (Philips, Amsterdam, Netherlands), equipped with an Android tablet (S4, Samsung, Seoul, Korea). LUS exams were collected prospectively during each participant's hospital visit at the bedside by study investigators, including both research coordinators and clinician-scientists. A standardized protocol that required minimal training was used for bedside image acquisition [23]; prior experience with performance of LUS ranged from none to well-experienced. Bilateral ultrasound scanning of 12 chest-wall areas (zones) was performed on participants in a supine, upright or semi-recumbent position. Each hemi-thorax was divided into 6 zones (upper and lower anterior, upper and lower lateral, upper and lower posterior) using the parasternal, anterior, and posterior axillary lines as anatomical landmarks (see S1 Fig). The Lumify system software "Lung" preset was selected with a default 12 cm image depth and gain of 36. Depending on the body habitus and age of the child, the depth could be adjusted by the operator to between 6 and 12 cm. Three-second video loops were acquired in each zone while holding the probe still in the sagittal plane (long axis) with the transducer marker pointed towards the participant's head. The acquired video loops were exported as .mp4 video files.

To develop the algorithm, positive consolidation videos were curated from participants enrolled with suspicion of pneumonia or associated complications. The clinical diagnosis of pneumonia was corroborated by the participant's discharge diagnosis supplemented with either CXR or CT scan findings (89% of participants) and respiratory viral panel results (77% of participants). Negative (no consolidation) videos were taken from the unaffected lung zones of participants with pneumonia as well as from participants without a diagnosis of pneumonia. Videos previously collected from adults with lower respiratory tract disease were added to enrich algorithm training. Clinical data to confirm the diagnosis of pneumonia was derived from a chart review conducted by research coordinators asynchronously from LUS image acquisition.

## Annotation

The annotation workflow (Fig 1), was a multi-step process with several quality control (QC) steps. Each participant's clinical data and lung ultrasound videos were reviewed by two physicians to confirm their merit for annotation. Annotation occurred at two levels: (*i*) at the entire video level for the lung features present; and (*ii*) at the frame level for localization of the lung features within each frame.

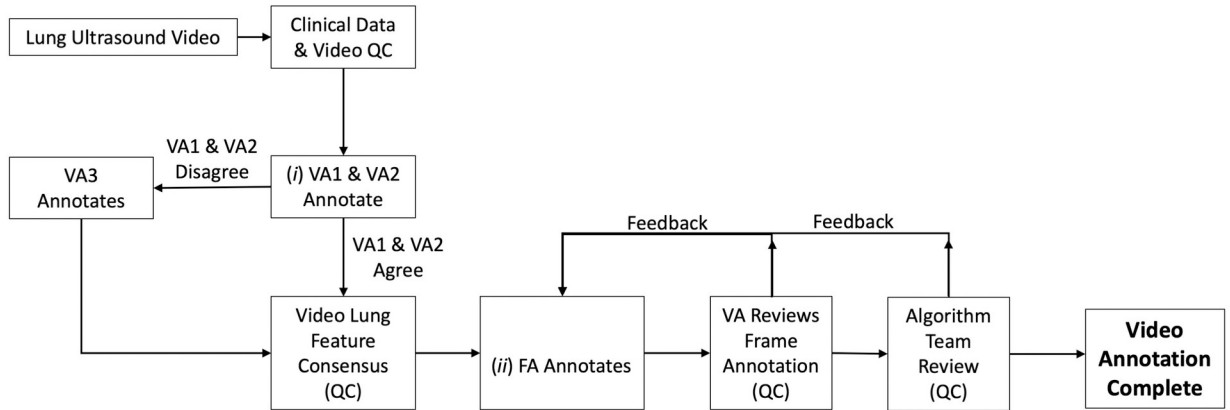

**Fig 1. Lung ultrasound annotation workflow.** Multi-step workflow for review and annotation of lung ultrasound videos is depicted. QC = quality control, VA = video annotator, FA = frame annotator.

**Video annotation.** The Lumify was operated at 20 frames/second. Every frame of every video containing relevant lung features was annotated. Video annotators (two physician LUS experts) labeled videos according to what features they exhibited using a standardized digital annotation platform. Labels included the normal features of non-diseased lungs (pleural line, A-lines) as well as potentially pathologic features of diseased lungs (consolidation, pleural effusion, sub-pleural changes, atelectasis, B-line, or merged B-line). Video annotators could reject a video for quality issues or exclusion criteria that may have been missed at the initial QC step. If the two annotators disagreed on the feature content, the video was passed to a third video annotator to arbitrate the video labeling. The arbitration of disagreements between video annotators was an iterative process. If a video was accepted, and if all annotators agreed on the feature content of the video, it was passed to frame annotators. For this study, only videos that contained consolidation confirmed by two video annotators were included in the positive consolidation set. The presence of air bronchograms was used to distinguish between consolidation resulting from infection (pneumonia), and suspected atelectasis, with the latter feature excluded from model training.

**Frame annotation.** Frame annotators had educational backgrounds in anatomy and physiology and were trained by physician LUS experts to recognize the specific LUS features of interest. Frame annotators assessed every frame of a video for lung features indicated by the video annotators and marked them with bounding boxes. Frames without any of the pre-defined pathologic features were included as examples of non-diseased (normal) areas of the lung. Frame annotation and bounding boxes were reviewed first by a video annotator and then verified by the algorithm team to ensure they were of sufficient quality and that the annotations were consistent and suitable for machine learning. Frame annotation was the most-labor intensive step of the process and was necessary to enable explainable outputs from the algorithm that localize features, distinguish between diseased and non-diseased tissues, and inspire clinician confidence in the algorithm.

## Machine learning protocol

**Data distribution.** The data were split into training, tuning, and holdout sets at the participant level. The training set was used to learn model weights, the tuning set was used to set model hyperparameters, and the holdout set was used to evaluate algorithm performance (see Table 1).

**Algorithm architecture.** The algorithm's architecture was designed to be capable of both video-level analysis and frame level localization. The primary goal was to identify whether a video was positive or negative for consolidation; additional goals were to pinpoint the exact frames within those videos, and the locations within those frames, where the consolidation was present. To achieve these dual requirements, a cascade architecture was adopted, meaning that the algorithm passed the input video through multiple processing steps.

**Table 1. Source and utilization of ultrasound data.**

| Subset | Hospitals | Participants | Exams | −Videos | +Videos |
|---|---|---|---|---|---|
| Training | 1 | 56 | 63 | 372 | 323 |
| Tuning | 2 | 24 | 24 | 71 | 122 |
| Holdout | 2 | 27 | 30 | 146 | 159 |
| Adult training | 6 | 44 | 102 | 843 | 498 |

−Videos = consolidation feature not present; +Videos = consolidation feature present

The first processing step was to classify each frame of the input video as either positive or negative for consolidation. The algorithm did this by employing a frame classifier that has a binary *yes/no* output to assess every frame within the video, categorizing them as either positive or negative for the presence of a consolidation. The convolutional neural network (CNN) architecture for the frame classifier was a VGG-like network trained on all frames in the training subset (using the frame labels, but ignoring bounding boxes that were drawn by annotators around the specific features of the consolidation) [24].

The second step was to classify the whole video as either exhibiting consolidation or not. Video classification was determined by applying a threshold to the number of positively-classified frames. For videos that were classified as positive in the second step, the algorithm executed a third processing step in which frames that were classified as positive were passed to an object detector to localize the consolidation with a bounding box. This third object detection step employed a single-shot architecture using a MobileNet V3 CNN as the base network [25, 26]. It was trained with the frame-level bounding boxes drawn by annotators around the consolidation in the training set. Once the model was trained, the total processing time of the three-step cascade on the ultrasound transducer's native tablet device was under ten seconds per video.

The net effect of the automated processing is to determine if a lung ultrasound video (taken from one of the different lung zones) exhibits a consolidation or not. When a video is labeled as showing a consolidation, the tablet replays the video with boxes drawn around the areas of consolidation that the algorithm identified in each frame. These boxes help the user to understand why the algorithm made its decision about a video classification. This creates transparency and builds trust in the automated algorithm. Users can then use their own medical expertise to confirm if the algorithm was accurate and make medical decisions.

## Sample size and analysis plan

The main study outcome was test performance of the algorithm at the video level to detect consolidation on lung ultrasound. Reference standard diagnosis of positive or negative pneumonia was derived from chart data confirming a clinical diagnosis of pneumonia along with consistent findings on radiologic imaging. Deep learning algorithms require a sufficient number of training samples to learn a viable model, but it is not always possible to predict in advance how many samples will be needed. Our goal was to achieve at least 90% accuracy which was considered on par with CXR as the current standard of care diagnostic imaging test for pneumonia. Model localization performance was analyzed using Intersection-over-Union (IoU) measuring the amount of overlap between the algorithm detected bounding boxes and expert-annotated bounding boxes outlining areas of consolidation. Diagnostic performance of the model was analyzed using Pearson Chi Square 2x2 contingency tables to calculate test characteristics (sensitivity, specificity, positive and negative predictive value) of the model in identifying videos with features of pulmonary consolidation as compared to our reference standard.

## Results

### Participant recruitment and characteristics

107 pediatric participants were enrolled, yielding 1,193 pediatric videos (approximately 10 videos per participant) for the algorithm development process. From these 1193 videos, 159 positive consolidation videos and 146 negative videos were utilized for algorithm testing (see Table 1). Low quality videos were excluded, with the main issues compromising quality related to poor transducer contact, bad transducer angle, transducer motion, or other organs

**Table 2. Population characteristics of pediatric participants contributing videos to algorithm development.**

| | |
|---|---|
| Median age in months, (IQR) | 72 (61.2) |
| Median weight in kilograms, (IQR) | 24.1 (19.35) |
| Positive respiratory pathogen test, n (%) | 57 (53%) |
| Influenza A | 4 |
| Influenza B | 6 |
| Human MPV | 12 |
| Rhinovirus/Enterovirus | 20 |
| Adenovirus | 3 |
| Parainfluenza | 4 |
| Respiratory syncytial virus | 6 |
| Mycoplasma pneumoniae | 6 |
| Bordetella Parapertussis | 1 |
| Coronavirus | 2 |
| Multiple pathogens* | 6 |
| Negative respiratory pathogen test, n | 28 (26%) |
| No respiratory pathogen testing completed, n | 22 (21%) |

*Specific frequency of each respiratory pathogen already displayed in table

obstructing visualization of the lungs. Median age of participants in the pediatric testing set was 6 years old (range 18 months to 17 years old, IQR = 61.2 months). LUS data from 44 adults hospitalized with pneumonia was added to the training data set. The addition of adult data served the dual purposes of increasing the amount of data (as deep learning networks require large amounts of training data) and adding more diversity. Specifically, the addition of adult data was felt to enhance representation of data from pediatric participants between 14–18 years old in the training data set. Table 2 shows additional characteristics of the pediatric population included in the algorithm testing.

Table 3 summarizes the performance metrics of the pediatric consolidation algorithm described above. Video level sensitivity was 88% and specificity was 89% for detecting consolidation. Average localization accuracy, as measured by the IoU between algorithm LUS detected and consolidation localized by expert assessment was 0.62.

## False negatives and positives

Of 159 positive videos, 19 were falsely classified by the algorithm as negative for consolidation. Over three quarters of the missed consolidations were due to the algorithm failing to detect

**Table 3. Two by two table illustrating sensitivity, specificity, positive predictive value, and negative predictive value of the pediatric consolidation feature algorithm to detect pneumonia.**

| | | Reference standard (ground truth) | | Sensitivity (95% CI) | 88% (82–93%) |
|---|---|---|---|---|---|
| | | Positive pneumonia | Negative pneumonia | Specificity (95% CI) | 89% (83–94%) |
| Ultrasound Algorithm | Positive consolidation | 140 | 16 | Positive Predictive Value (95% CI) | 89% (85–93%) |
| | Negative consolidation | 19 | 130 | Negative Predictive Value (95% CI) | 87% (82–91%) |

CI = confidence interval

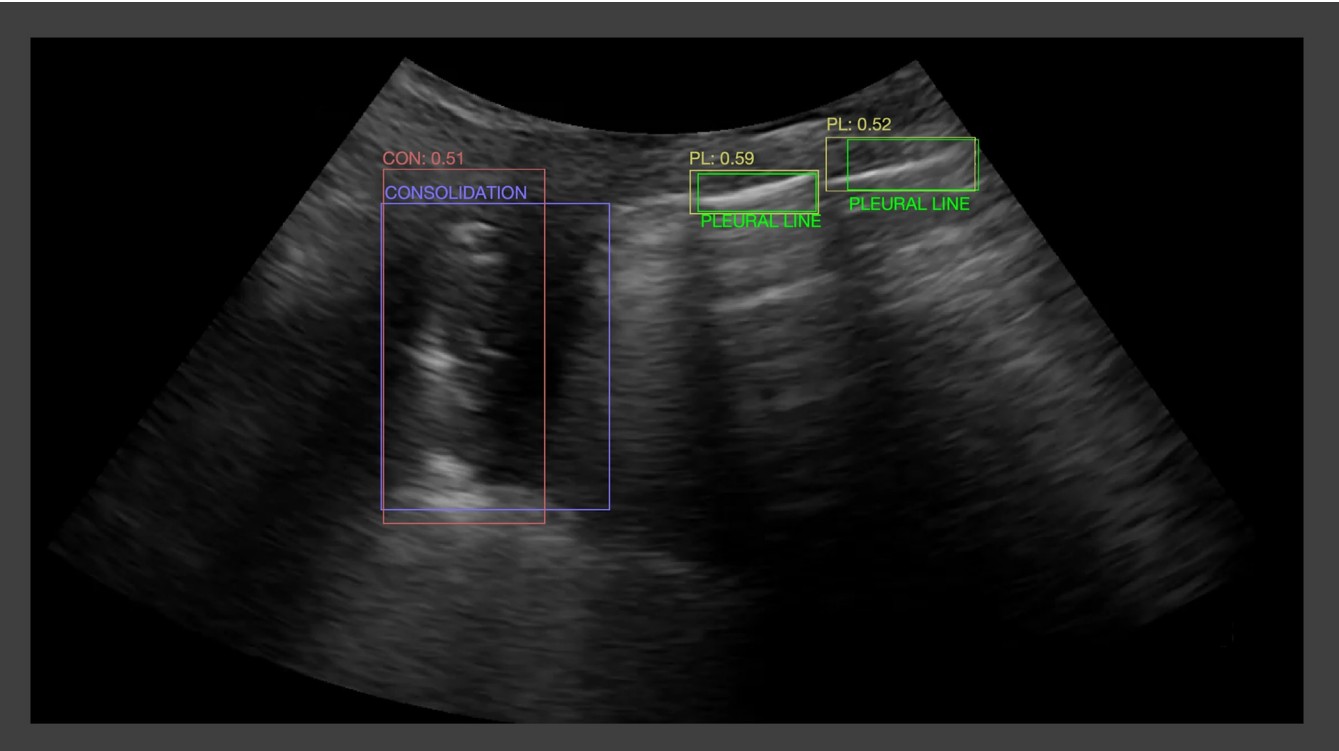

**Fig 2. Sample algorithm output for consolidation and normal pleural line features overlaid on expert-defined bounding boxes for the same features.**

transiently visible consolidations that moved behind a rib shadow intermittently as the participant respires. The remaining missed consolidations were smaller in size (<5 mm), at the boundary between actionable consolidations and clinically insignificant ones [16].

Of 146 negative videos, 16 were falsely classified by the algorithm as positive for a consolidation. Over half of the false positives (9/16) were triggered by abnormal pleural lines that were either thickened, irregular, or interrupted. The remaining false positives were triggered by atelectasis or B-lines emanating from a thickened or irregular pleural line.

An example of the algorithm output overlying the consolidation localized by conventional CXR, can be seen in Fig 2.

Red and yellow rectangles indicate the algorithm output for detecting consolidation and normal pleural line features, respectively, with the numbers above the rectangles representing the algorithm output confidence scores. Purple and green rectangles indicate the expert annotated bounding boxes for consolidation and normal pleural line features, respectively.

## Discussion

Severe pneumonia remains the leading cause of death among young children worldwide [1]. Current international guidelines that base diagnosis of pneumonia on history and clinical exam are intended to identify young children who would benefit from empiric antibiotics and do not offer high diagnostic accuracy [20, 27]. This inaccuracy places young children at risk for a missed diagnosis of bacterial pneumonia and its complications such as empyema, while also increasing exposure to inappropriate antibiotics among children with viral upper-respiratory tract infections and/or bronchiolitis that mimic the clinical presentation of bacterial pneumonia [6]. LUS has emerged as a highly accurate imaging modality to identify pulmonary

consolidation as well as complications of severe pneumonia, such as pleural effusions and empyema, in well-resourced settings [12–16, 20]. However, in low resource settings where the need for improved diagnostics is most urgent, equipment to perform lung ultrasound (and even conventional CXR) and the expertise to interpret imaging are rarely available. To address this need, we developed and tested a point-of-care algorithm that detects LUS features of pulmonary consolidation in children. Our results demonstrated promising diagnostic accuracy for this feature interpretation algorithm, with performance near expert level of 90% sensitivity and specificity. We also demonstrated a strong IoU (0.62) that corresponds to roughly 79% linear dimension overlap between the algorithm's feature detection and expert annotation. This provides a powerful proof-of-concept that an AI algorithm applied to point-of-care LUS videos can be deployed to improve diagnosis of pneumonia in children, and may be particularly impactful in low resource settings.

There has been a tremendous expansion of research into the development and testing of AI to diagnose lung features [21, 22, 28, 29]. However, ours is one of the only studies validated on a point-of-care device specifically detecting features of pediatric pneumonia across entire videos. Correa *et al.* developed a model for analysis of pediatric LUS videos based on a neural network classifying small feature vectors that are distilled from a few frames of each video [30]. They achieved a sensitivity of 90.9% and specificity of 100% to correctly identify regions of ultrasound that contained a consolidation. However, the combined design and test dataset was very small: the total number of participants was 21, and the total number of video frames was 60. The small dataset precluded the ability to compute reliable *patient-level* performance metrics—which is what matters most for clinical applications. Furthermore, the images were acquired using an Ultrasonix device, (BK Medical, Vancouver, British Columbia, Canada), which is a larger and more expensive device, thus limiting generalizability to low-resource settings where such equipment would not be available. In contrast, our algorithm was trained on videos acquired on an ultraportable device by minimally trained research personnel following a simple standard protocol, making our findings more transferable to austere environments and a variety of operators. Importantly, all images were acquired point-of-care at the bedside by research team members using a study-specific standardized operating procedure; images were not acquired by radiologists or professional sonographers. In addition, rather than simply detecting a region of interest, our algorithm directly identified consolidations making it more useful for novice operators. Nti *et al.* explored the use of AI software to help guide novices in identifying lung features acquired on a Zonare machine (Zs3, Mindray North America, Mahwah, NJ, USA) and found it helped improve novice recognition and diagnosis of abnormal features [31]. They did not employ an ultraportable device, however their pilot study demonstrated the potential for scalability of software that is agnostic to specific machines or companies.

## Limitations

Our accuracy fell short of our 90% goal which may be attributable to the relatively small training set size. One of the biggest challenges for our algorithm was accurately diagnosing transient features. This is mainly due to the current version of the video classifier being frame-based and not accounting for temporal dynamics, (*i.e.*, a consolidation that disappears behind the rib with each respiratory cycle). Future iterations of the algorithm will incorporate methods that model temporal dynamics and should be able to detect a better portion of these false negatives. The algorithm was developed, trained, and tested only on images with consolidations caused by pneumonia. In addition, there are other diseases that can lead to alveolar injury and filling seen on LUS. Therefore, it is not clear how well this algorithm will be able to

distinguish pneumonia from other causes of consolidation artifacts on LUS (*e.g.*, atelectasis). Given the higher prevalence in adult populations of these other illnesses, the addition of adult training data is a potential contributing factor to this as well. Our study did not test participants systematically for viral or bacterial pathogens, and thus we were unable to probe relationships between the causative organism and findings on LUS. Our study design also did not allow us to determine if our LUS imaging protocol and algorithm could identify small consolidations (< 5mm), or the relationship between specific features of consolidations (such as dimensions) and etiology, as has been suggested by a prior study [32]. Since pneumonia is a common cause of lower respiratory infection in low-resource outpatient settings, the algorithm's specificity may be adequate for this patient population, but limited in other settings. Finally, the algorithm was highly unpredictable in discerning ambiguous or borderline lung features, for example small consolidations or irregular pleural lines. These borderline features can also be difficult for experts to classify and there remains ambiguity as to the clinical significance for borderline features, such as sub-centimeter consolidations [33–35]. Ambiguous features are a general problem in the application of AI to medicine, one which begs for a principled and robust approach that will alert the user when the algorithm is unsure of its findings. Future directions include exploring the use of multiclass networks and networks that provide an auxiliary output indicating the uncertainty of their findings as potential avenues for ameliorating the impact of ambiguous patterns. Wearable technology may also provide the opportunity to acquire continuous serial imaging data over an entire disease [36].

## Conclusion

A deep learning algorithm developed from images acquired on an ultraportable device demonstrates high accuracy, sensitivity, and specificity, for identification of consolidation features on pediatric chest ultrasound among children with pneumonia. The capacity to deploy automated and accurate diagnostic support on an ultraportable point-of-care device has important implications for global health, especially in low-resource or austere settings.

## Supporting information

**S1 Fig. Scanning zones for pediatric lung ultrasound.** Lung image acquisition areas. Anterior scan areas: right hemithorax–areas 1, 2, 3, and 4; left hemithorax–areas 5, 6, 7, and 8. Posterior scan areas: right hemithorax–areas 9 and 10; left hemithorax–areas 11 and 12. (Created with BioRender.com).
(PDF)

## Acknowledgments

The authors thank Erin Merrifield (OHSU) for her help with participant enrollment, ultrasound image acquisition, and clinical data organization. They also thank the OHSU Center for Regenerative Medicine staff members Amber Halse, Andrew Jones, Jack Lazar, Yuan Zhang, Annie Cao, and Katelyn Hostetler for their frame annotation of ultrasound images.

## Author Contributions

**Conceptualization:** Cynthia R. Gregory, Marc Vindas.

**Data curation:** Meihua Zhu, Cynthia R. Gregory, Courosh Mehanian, Jailyn Avila, Nick Avitable, Di Coneybeare, Devjani Das, Almaz Dessie, Thomas M. Kennedy, Joni Rabiner, Laurie Malia, Lorraine Ng, Megan Nye, Peter Weimersheimer, Kenton Gregory.

**Formal analysis:** Cynthia R. Gregory, Courosh Mehanian, Sourabh Kulhare, Rachel Millin, Xinliang Zheng.

**Funding acquisition:** Kenton Gregory.

**Investigation:** David Kessler, Christina Lancioni.

**Methodology:** Meihua Zhu, Cynthia R. Gregory, Courosh Mehanian, Sourabh Kulhare, Kenton Gregory, Xinliang Zheng, Matthew P. Horning, Mike Stone, Fen Wang.

**Project administration:** Cynthia R. Gregory, Megan Nye, Marc Vindas, Kenton Gregory.

**Supervision:** David Kessler, Cynthia R. Gregory, Kenton Gregory, Christina Lancioni.

**Writing – original draft:** David Kessler, Courosh Mehanian.

**Writing – review & editing:** David Kessler, Meihua Zhu, Cynthia R. Gregory, Courosh Mehanian, Jailyn Avila, Nick Avitable, Di Coneybeare, Devjani Das, Almaz Dessie, Thomas M. Kennedy, Joni Rabiner, Laurie Malia, Lorraine Ng, Megan Nye, Marc Vindas, Peter Weimersheimer, Kenton Gregory, Christina Lancioni.

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
