## [Decision Letter · Decision Letter 0]

15 Feb 2024

PONE-D-23-38634Development and testing of a deep learning algorithm to detect pediatric pneumonia on a portable ultrasound devicePLOS ONE

Dear Dr. Kessler,

Thank you for submitting your manuscript to PLOS ONE. After careful consideration, we feel that it has merit but does not fully meet PLOS ONE’s publication criteria as it currently stands. Therefore, we invite you to submit a revised version of the manuscript that addresses the points raised during the review process.

We look forward to receiving your revised manuscript.

Kind regards,

Tai-Heng Chen, M.D., Ph.D.

Academic Editor

PLOS ONE

This work was supported via a Defense Advanced Research Projects Agency award “Hand-Held Convolutional-Neural-Network based Field Diagnostic Ultrasound” Technology Investment Agreement No. HR0011-17-3-0001 to Inventive Government Solutions, LLC and subcontracted to OHSU.  Co-funding was provided by the Global Good Fund, under auspices of the Bill and Melinda Gates Foundation Trust, awarded to Intellectual Ventures Laboratory. Dr. Kenton Gregory received salary support thru Intellectual Ventures Laboratory, a company that may have a commercial interest in the results of this research and technology.  

I have read the journal's policy and the authors of this manuscript have the following competing interests: Drs. Kenton and Cynthia Gregory received salary support thru Intellectual Ventures Laboratory, a company that may have a commercial interest in the results of this research and technology.  This potential conflict for Drs. Kenton and Cynthia Gregory has been reviewed and managed by OHSU.

We note that one or more of the authors are employed by a commercial company.

“The funder provided support in the form of salaries for authors, but did not have any additional role in the study design, data collection and analysis, decision to publish, or preparation of the manuscript. The specific roles of these authors are articulated in the ‘author contributions’ section.”

5. In the online submission form, you indicated that the data underlying the results presented in the study are available upon request from the corresponding author.

6. One of the noted authors is a group or consortium. In addition to naming the author group, please list the individual authors and affiliations within this group in the acknowledgments section of your manuscript. Please also indicate clearly a lead author for this group along with a contact email address.

Reviewers' comments:

Reviewer's Responses to Questions

**Comments to the Author**

1. Is the manuscript technically sound, and do the data support the conclusions?

Reviewer #1: Yes

Reviewer #2: Yes

2. Has the statistical analysis been performed appropriately and rigorously? 

Reviewer #1: Yes

Reviewer #2: I Don't Know

3. Have the authors made all data underlying the findings in their manuscript fully available?

Reviewer #1: Yes

Reviewer #2: Yes

4. Is the manuscript presented in an intelligible fashion and written in standard English?

Reviewer #1: Yes

Reviewer #2: Yes

5. Review Comments to the Author

Reviewer #1: Thank you very much for the opportunity to review the manuscript. I found it very well-written, clearly written, enjoyable to read, and worthy of publication. I share the vision the authors have put forth and believe this is a very important work to improve access to imaging. The results are also of great interest to the scientific community.

I had a few minor comments that the authors may consider when submitting any revisions.

-Line 58, I agree CXR is the current standard of care, but in meta-analysis lung ultrasound seems to perform better. Perhaps a distinction could be made in this regard.

-The authors mention the use of adult imaging. I understand the focus of your manuscript is on pediatric patients. If possible, I still think the adults you scanned should ideally be included in the description of the study population section. Were adults enrolled with the same criteria?

-Can you please consider explaining the rationale for adding adult images in a little more detail? You state it is to “enrich” the data but I think it may be beneficial to explain what about the adult data enriches the AI – for example more studies to train the AI, more generalizability etc.? Your manuscript specifically focuses on pediatric groups but of course this could also be used in adults. I think it could also be an interesting area of discussion in any differences between the adult and pediatric populations as a future consideration but I wouldn’t want this to distract from the focus of your manuscript which I feel is already well written as it is. Perhaps a few sentences could be included commenting on this in the discussion if you feel it could be beneficial. Otherwise, I also understand if you want to keep the focus of the manuscript on children and not comment further.

-If you had 117 unique exams, multiplied by 12 that would be 1,404 video clips. You report using 1,193 video clips. I assume the 211 difference related to clips that were not of adequate quality? What were some of the quality issues that made a clip not usable? For example, were their problems with motion artifact or not enough gel? If you could provide exact data regarding clips were rejected that would be interesting if you happened to record it, but I think even just some generalized qualitative description of things that affected quality would be helpful.

-Were the images acquired point-of-care at the bedside or in an ultrasound suite or a combination? Were the exams performed by physicians or by sonographers or a combination? Readers may also benefit from a little more information about how the people acquiring the images were trained and what their background was prior to the study. In the discussion you describe “minimally trained research personnel” line 306. I assume the multi-center nature of your study may have individuals with a variety of backgrounds acquiring the imaging? More information about those scanning and how many people scanned would be helpful/interesting if possible.

-How many frames per second does Lumify record? Was every single frame labeled?

-I found the discussion section enlightening, relevant, and useful. I particularly appreciate the discussion on the challenges of AI in relation to ambiguous cases and feel the authors have provided a very useful discussion of the promise of this work and areas of further optimization going forward.

-My copy of Figure 1 and Figure 2 are grainy. I’m not sure if this is related to the reviewer quality shared with me but please double check the image submitted before publication to ensure the image and text is as high resolution possible.

-Again, I commend the authors on this work. I believe they have shown a very strong proof of concept for this approach and their work merits publication.

Reviewer #2: The paper subject is very interesting, as lung ultrasound is a reliable method for lung assessment in children respiratory pathology and the study background accordingly to premises.

There are limitations that must be acknowledge, like retro scapular regions, difficult to evaluate

Please insert data to clarify the following issues:

-pneumonia diagnosis criteria used for this study

-consolidations might be present even in viral pneumonia, please state the relation with etiology

-no clear data on relation between dimensions and etiology, even if several paper suggests the relation( eg.Kharasch S, Duggan NM, Cohen AR, Shokoohi H. Lung Ultrasound in Children with Respiratory Tract Infections: Viral, Bacterial or COVID-19? A Narrative Review. Open Access Emerg Med. 2020, Buonsenso D, Musolino A, Ferro V, et al. Role of lung ultrasound for the etiological diagnosis of acute lower respiratory tract infection (ALRTI) in children: a prospective study , J Ultrasound. 2021)

-please provide the explanation for the comparison method, as mention or" CXR either CT scan , there is mentioned that :"Reference standard diagnosis of positive or negative pneumonia was derived from chart data confirming a clinical diagnosis of pneumonia along with consistent findings on radiologic imaging"

-as pneumonia is not just an imaging diagnosis, maybe the change title should be considered because the study seemed to evaluated just the presence of consolidations

-the study declares that movies with consolidations were evaluated, no data on dimensions, and would be a significant bias, as they suggest etiology

-small consolidations < 1-1.5 cm are not visible on CXR(considered as gold standard) Berce V, Tomazin M, Gorenjak M, Berce T, Lovrenčič B. The usefulness of lung ultrasound for the aetiological diagnosis of community-acquired pneumonia in children. Sci Rep. 2019, therefore the US could be more sensitive in detecting small lesions, please comment on that, how was this considered in your study

-No data on pleural effusion

-No data on empiema or B lines, nor on atelectasis

6. PLOS authors have the option to publish the peer review history of their article (what does this mean?). If published, this will include your full peer review and any attached files.

Reviewer #1: **Yes: **Thomas J. Marini

Reviewer #2: No

---

## [Author Response · Author response to Decision Letter 0]

29 Apr 2024

Response to Reviewers:

Re: PONE-D-23-38634, “Development and testing of a deep learning algorithm to detect pediatric pneumonia on a portable ultrasound device”

Reviewer #1:

1) I agree CXR is the current standard of care, but in meta-analysis lung ultrasound seems to perform better. Perhaps a distinction could be made in this regard:

 We agree that recent meta-analysis comparing lung ultrasound (LUS) to CXR have demonstrated robust diagnostic accuracy in both pediatric and adult populations and have added additional comment and references in the revised Introduction.

2) The authors mention the use of adult imaging. I understand the focus of your manuscript is on pediatric patients. If possible, I still think the adults you scanned should ideally be included in the description of the study population section. Were adults enrolled with the same criteria? 

 Adults 18-80 years old with a clinical diagnosis of pneumonia and CXR and/or chest CT evidence of consolidation were enrolled separately, following provision of written informed consent. Adult patients were excluded if they were intubated, had active sepsis, cardiogenic pulmonary edema, known lung cancer, pulmonary embolus, chronic bronchiectasis or cystic fibrosis, or could not tolerate the LUS procedure. These details have been added to the Methods section of the revised manuscript. 

3) Please consider explaining the rationale for adding adult images in a little more detail. You state it is to “enrich” the data but I think it may be beneficial to explain what about the adult data enriches the AI – for example more studies to train the AI, more generalizability etc.? …I think it could also be an interesting area of discussion in any differences between the adult and pediatric populations as a future consideration but I wouldn’t want this to distract from the focus of your manuscript which I feel is already well written as it is. Perhaps a few sentences could be included commenting on this in the discussion if you feel it could be beneficial. Otherwise, I also understand if you want to keep the focus of the manuscript on children and not comment further.

 We added to the Results section of the revised manuscript additional detail regarding the rational for including in the training data set LUS data from 44 adults hospitalized with pneumonia as follows: “The addition of adult data served the dual purposes of increasing the amount of data (as deep learning networks require large amounts of training data) and adding more diversity. Specifically, the addition of adult data was felt to enhance representation of data from pediatric patients between 14-18 years old in the training data set.”

 In our analysis, we did not specifically compare LUS imaging findings between pediatric and adult participants with consolidations. Thus, we feel that adding an additional discussion on age-based differences in identification of consolidations by LUS would not be appropriate for this manuscript.

4) If you had 117 unique exams, multiplied by 12 that would be 1,404 video clips. You report using 1,193 video clips. I assume the 211 difference related to clips that were not of adequate quality? What were some of the quality issues that made a clip not usable? For example, were their problems with motion artifact or not enough gel? If you could provide exact data regarding clips were rejected that would be interesting if you happened to record it, but I think even just some generalized qualitative description of things that affected quality would be helpful. 

 There were 117 unique exams, but the number of videos per exam was quite variable, ranging from 1 to 38. The average number of videos per exam was 10.2. Low quality videos were excluded, with the main issues compromising quality related to poor transducer contact, bad transducer angle, transducer motion, or other organs obstructing visualization of the lung. An explanation for exclusion of low quality videos is provided in the Results section of the revised manuscript. 

5) Were the images acquired point-of-care at the bedside or in an ultrasound suite or a combination? Were the exams performed by physicians or by sonographers or a combination? Readers may also benefit from a little more information about how the people acquiring the images were trained and what their background was prior to the study. In the discussion you describe “minimally trained research personnel” line 306. I assume the multi-center nature of your study may have individuals with a variety of backgrounds acquiring the imaging? More information about those scanning and how many people scanned would be helpful/interesting if possible.

 All images were acquired point-of-care at the bedside by clinicians or research staff members following training using a study-specific standardized operating procedure; images were not acquired by radiologists or professional sonographers. Prior experience with performance of lung ultrasonography ranged from none to moderate-level of experience. At one pediatric recruitment site, all images were acquired by a single clinician-researcher; at our second pediatric recruitment site, images were acquired by multiple clinician-researchers. Additional details regarding performance of the LUS procedure are included in the Methods section of the revised manuscript. 

6) How many frames per second does Lumify record? Was every single frame labeled?

The Lumify was operated at 20 frames/second. Every frame of every video containing relevant lung features was annotated. These details are included in the Methods section of the revised manuscript. 

7) My copy of Figure 1 and Figure 2 are grainy. 

 We apologize for the poor quality of the uploaded images. We have improved the quality of our figures for the revised manuscript. 

Reviewer #2:

1) Please insert data to clarify the pneumonia diagnosis criteria used for this study

 Following our approved IRB protocols and after receiving written informed consent and assent (if applicable), the medical records from each participant’s relevant hospitalization were reviewed to identify individuals diagnosed with pneumonia. Specifically, the following details have been added to the Methods section of the revised manuscript. “The medical records from each participant’s relevant hospitalization were reviewed to obtain demographic and clinical data and to identify individuals diagnosed with pneumonia. Specifically, data detailing admission and discharge diagnoses, results of any microbiologic testing (e.g., respiratory viral panel results were available for 77% of subjects), performance of pulmonary procedures (eg, chest tube placement; bronchoscopy), and images and clinical radiologists’ interpretation of chest imaging (available for 89% of subjects), were collected.” 

2) Consolidations might be present even in viral pneumonia, please state the relation with etiology

 We agree with the Reviewer that viral infections can be associated with focal pulmonary consolidations. While 77% of the subjects were tested for viral pathogens, our study was not designed or powered to examine correlations between identification of viral pathogens and focal lung consolidations, and we were unfortunately not able to probe this relationship in our data set. We have included this as a limitation in our revised manuscript.

3) No clear data on relation between dimensions and etiology, even if several paper suggests the relation

 We appreciate the Reviewer’s reference to prior studies demonstrating the capacity of specific LUS findings to delineate different microbiologic (eg viral vs bacterial) causes of lower respiratory tract infection in children. As explained in our response above (Reviewer 2, #2), our study did not include comprehensive testing for causative pathogens and we are unfortunately unable to perform such an analysis with our current data set.

4) Please provide the explanation for the comparison method, as mentions either CXR or CT scan; there is mentioned that :"Reference standard diagnosis of positive or negative pneumonia was derived from chart data confirming a clinical diagnosis of pneumonia along with consistent findings on radiologic imaging"

 The majority (i.e., 89%) of pediatric participants had CXRs/chest CTs obtained as part of their clinical care. We reviewed their interpretation to ensure that children with a clinical diagnosis of pneumonia had evidence of consolidation on routine clinical imaging.

5) As pneumonia is not just an imaging diagnosis, maybe the change title should be considered because the study seemed to evaluated just the presence of consolidations

 We agree with the Reviewer that pneumonia is a clinical diagnosis that can be supported by findings of consolidation on lung imaging, and appreciate the suggestion to refine the title of this manuscript. We have retitle the revised manuscript “Development and testing of a deep learning algorithm to detect lung consolidation among children with pneumonia using hand-held ultrasound.”

6) The study declares that movies with consolidations were evaluated, no data on dimensions, and would be a significant bias, as they suggest etiology

 As detailed above (Reviewer 2, #2 and #3 above), our study was not designed to identify or distinguish the etiology of pneumonia among participating children and this is now included as a study limitation in the revised manuscript. 

7) Small consolidations < 1-1.5 cm are not visible on CXR (considered as gold standard) Berce V, Tomazin M, Gorenjak M, Berce T, Lovrenčič B. The usefulness of lung ultrasound for the aetiological diagnosis of community-acquired pneumonia in children. Sci Rep. 2019. Therefore, the US could be more sensitive in detecting small lesions, please comment on that, how was this considered in your study

We agree that US may be more sensitive than CXR for detection of small consolidations, and this was a consideration in regard to the algorithm’s false negative and false positive classifications. As discussed in the results section, a quarter of the consolidations missed by the algorithm were smaller in size, while over half of the false positives were triggered by abnormal pleural lines. Our study design, built for initial algorithm development and validation, did not allow us to examine the sensitivity of LUS for small consolidations less than 5 mm. We look forward to performing a study that examines the test characteristics of our LUS protocol and algorithm to address this important consideration. We have, however, added into the Discussion of the revised manuscript the potential for LUS (including the reference from Berce et al) to identify small consolidations. 

8) No data on pleural effusion and no data on empyema or B lines, nor on atelectasis

 The primary goal of this project was to develop an AI algorithm to detect lung consolidations, specifically, among children with a clinical diagnosis of pneumonia using a hand-held ultrasound devise. The dominant lung ultrasound features among this pediatric population were pleural lines, A-lines, B-lines, sub-pleural consolidations and consolidations of different sizes. In this cohort, pleural effusions were present in around 20% of the annotated videos; however, there were insufficient effusion data to allow development of the algorithm for detection of pleural effusions in children. The presence of empyema, B lines, and suspected atelectasis were not formally analyzed.

---

## [Decision Letter · Decision Letter 1]

23 Jun 2024

PONE-D-23-38634R1Development and testing of a deep learning algorithm to detect lung consolidation among children with pneumonia using hand-held ultrasoundPLOS ONE

Dear Dr. Kessler,

Thank you for submitting your manuscript to PLOS ONE. After careful consideration, we feel that it has merit but does not fully meet PLOS ONE’s publication criteria as it currently stands. Therefore, we invite you to submit a revised version of the manuscript that addresses the points raised during the review process.

We look forward to receiving your revised manuscript.

Kind regards,

Tai-Heng Chen, M.D., Ph.D.

Academic Editor

PLOS ONE

Reviewers' comments:

Reviewer's Responses to Questions

**Comments to the Author**

1. If the authors have adequately addressed your comments raised in a previous round of review and you feel that this manuscript is now acceptable for publication, you may indicate that here to bypass the “Comments to the Author” section, enter your conflict of interest statement in the “Confidential to Editor” section, and submit your "Accept" recommendation.

Reviewer #1: All comments have been addressed

Reviewer #2: All comments have been addressed

2. Is the manuscript technically sound, and do the data support the conclusions?

Reviewer #1: Yes

Reviewer #2: Yes

3. Has the statistical analysis been performed appropriately and rigorously? 

Reviewer #1: Yes

Reviewer #2: Yes

4. Have the authors made all data underlying the findings in their manuscript fully available?

Reviewer #1: Yes

Reviewer #2: Yes

5. Is the manuscript presented in an intelligible fashion and written in standard English?

Reviewer #1: Yes

Reviewer #2: Yes

6. Review Comments to the Author

Reviewer #1: Thank you for your revisions and work on this manuscript. The original comments from the review have been addressed.

Reviewer #2: This is an interesting study on AI effect regarding lung consolidation detection in children pneumonia. Please provide data on pneumonia consolidation differentiation from atelectasis , as bronchogram exists in pneumonia and no in atelectasis. This issue must be addressed, in order not to create confusion.

Please provide diagnosis criteria for pneumonia used for the study; there is an addendum: "“The medical records from each participant’s relevant

hospitalization were reviewed to obtain demographic and clinical data and to identify

individuals diagnosed with pneumonia. Specifically, data detailing admission and

discharge diagnoses, results of any microbiologic testing.." but is not clear how did the patients were selected by protocol and criteria used. It looks like diagnosis was made randomly by other physicians, but what were the criteria used; were not established in the study design

7. PLOS authors have the option to publish the peer review history of their article (what does this mean?). If published, this will include your full peer review and any attached files.

Reviewer #1: **Yes: **Thomas Marini

Reviewer #2: **Yes: **Ioana Ciuca

---

## [Author Response · Author response to Decision Letter 1]

16 Jul 2024

Response to Reviewers 

PONE-D-23-38634R1

Development and testing of a deep learning algorithm to detect lung consolidation among children with pneumonia using hand-held ultrasound

Review Comments to the Author

Reviewer #1: 

1. Thank you for your revisions and work on this manuscript. The original comments from the review have been addressed.

Thank you for the valuable feedback and helping to improve our manuscript.

Reviewer #2: 

2. This is an interesting study on AI effect regarding lung consolidation detection in children pneumonia. Please provide data on pneumonia consolidation differentiation from atelectasis , as bronchogram exists in pneumonia and no in atelectasis. This issue must be addressed, in order not to create confusion.

We thank the reviewer for highlighting that consolidation due to pneumonia must be distinguished from atelectasis, and that the presence of dynamic air bronchograms supports a finding of consolidation resulting from pneumonia. For this study, all ultrasound images were reviewed and independently labelled by two physician experts in lung ultrasound. Reviewers separately annotated features of consolidation and atelectasis. The presence of air bronchograms was systematically used to distinguish consolidation resulting from suspected infection (pneumonia), from suspected atelectasis, and images annotated accordingly. For this study only videos that contained consolidation confirmed by two video annotators were included in the positive consolidation set. Suspected atelectasis was not included as a feature of consolidation reflecting pneumonia in training or testing of the model. 

We have discussed this in the following sections of the manuscript:

Methods: (line 172) LUS images were annotated for atelectasis

“Labels included…atelectasis…”

Methods: (line 178): 

“For this study, only videos that contained consolidation confirmed by two video annotators were included in the positive consolidation set. The presence of air bronchograms was routinely used to distinguish consolidation resulting from suspected infection (pneumonia) from suspected atelectasis with the latter feature deliberately excluded from model training.” 

Results: (line 294) Some false positives were attributed to atelectasis

“…the remaining false positives were triggered by atelectasis…”

Limitations (line 362): “…it is not clear how well this algorithm will be able to distinguish pneumonia from other causes of consolidation artifacts on LUS (e.g. atelectasis)

3. Please provide diagnosis criteria for pneumonia used for the study; there is an addendum: "“The medical records from each participant’s relevant

hospitalization were reviewed to obtain demographic and clinical data and to identify

individuals diagnosed with pneumonia. Specifically, data detailing admission and

discharge diagnoses, results of any microbiologic testing.." but is not clear how did the patients were selected by protocol and criteria used. It looks like diagnosis was made randomly by other physicians, but what were the criteria used; were not established in the study design

To identify potentially eligible patients for study inclusion, emergency room and hospital ward censuses were reviewed by the study team to identify individuals being evaluated or admitted with suspected pneumonia who met study eligibility requirements, per approved IRB protocols. Following receipt of informed consent (and assent, if appropriate) from eligible participants, LUS was performed. Subsequently, data from the medical records from each participant’s relevant hospitalization were reviewed to obtain demographic and clinical data and to identify individuals diagnosed and treated for pneumonia. Specifically, data detailing admission and discharge diagnoses, results of any microbiologic testing (respiratory viral panel results were available for 77% of subjects), treatment course, performance of pulmonary procedures (eg, chest tube placement; bronchoscopy), and clinical radiologists’ interpretation of chest imaging (available for 89% of subjects), were collected. 

Using this approach, participants with diagnostic studies, clinical course, and documentation supporting a diagnosis of pneumonia were systematically delineated from participants with other disease processes who were not diagnosed with pneumonia (see “Scanning protocol and imaging data” for further details). 

As the reviewer notes, the diagnosis of pneumonia is a clinical diagnosis that can include signs such as fever, cough, pleuritic chest pain, and exam findings may include crackles on lung exam with varying degrees of respiratory distress (e.g. tachypnea and/or retractions). 

Therefore, our primary definition for pneumonia relied upon our extensive chart review to confirm that a clinical diagnosis of pneumonia was made by treating physicians as evidenced by ICD diagnostic codes for admission and/or discharge diagnosis of pneumonia. We have included further details of this process outline above with edits to the methods in this revision of this manuscript.

---

## [Decision Letter · Decision Letter 2]

6 Aug 2024

Development and testing of a deep learning algorithm to detect lung consolidation among children with pneumonia using hand-held ultrasound

PONE-D-23-38634R2

Dear Dr. Kessler,

We’re pleased to inform you that your manuscript has been judged scientifically suitable for publication and will be formally accepted for publication once it meets all outstanding technical requirements.

Kind regards,

Tai-Heng Chen, M.D., Ph.D.

Academic Editor

PLOS ONE

Additional Editor Comments (optional):

Reviewers' comments:

Reviewer's Responses to Questions

**Comments to the Author**

1. If the authors have adequately addressed your comments raised in a previous round of review and you feel that this manuscript is now acceptable for publication, you may indicate that here to bypass the “Comments to the Author” section, enter your conflict of interest statement in the “Confidential to Editor” section, and submit your "Accept" recommendation.

Reviewer #1: All comments have been addressed

2. Is the manuscript technically sound, and do the data support the conclusions?

Reviewer #1: Yes

3. Has the statistical analysis been performed appropriately and rigorously? 

Reviewer #1: Yes

4. Have the authors made all data underlying the findings in their manuscript fully available?

Reviewer #1: Yes

5. Is the manuscript presented in an intelligible fashion and written in standard English?

Reviewer #1: Yes

6. Review Comments to the Author

Reviewer #1: (No Response)

7. PLOS authors have the option to publish the peer review history of their article (what does this mean?). If published, this will include your full peer review and any attached files.

Reviewer #1: **Yes: **Thomas Marini

---

## [Editor Report · Acceptance letter]

15 Aug 2024

PONE-D-23-38634R2 

PLOS ONE

Dear Dr. Kessler, 

I'm pleased to inform you that your manuscript has been deemed suitable for publication in PLOS ONE. Congratulations! Your manuscript is now being handed over to our production team.

Kind regards, 

on behalf of

Dr. Tai-Heng Chen 

Academic Editor

PLOS ONE